# Actively Mode Locked Raman Fiber Laser with Multimode LD Pumping

Alexey G. Kuznetsov [1,*], Sergey I. Kablukov [1], Yuri A. Timirtdinov [1] and Sergey A. Babin [1,2]

[1] Institute of Automation and Electrometry, Siberian Branch of the Russian Academy of Sciences, 1 Ac. Koptyug Ave., 630090 Novosibirsk, Russia; kab@iae.nsk.su (S.I.K.); timirtdinov@iae.nsk.su (Y.A.T.); babin@iae.nsk.su (S.A.B.)

[2] Physics Department, Novosibirsk State University, 2 Pirogova Str., 630090 Novosibirsk, Russia

[*] Correspondence: kuznetsovag@iae.nsk.su

**Abstract:** We present our recent experimental results on the pulsed regimes of Raman conversion of highly multimode laser diode (LD) pump radiation into the 1st and higher order Stokes radiation in multimode graded-index fibers. Three different linear cavities of Raman fiber laser with the modulation of losses (by acousto-optic modulator, AOM) or gain (by LD current) are explored and compared. An LD with wavelength of 976 nm is used for pumping enabling Raman lasing at wavelength of the 1st (1018 nm) and 2nd (1064 nm) Stokes orders. At ~27.2-kHz repetition rate corresponding to the laser cavity round-trip frequency (i.e., in the mode-locking regime), nanosecond pulses have been observed for both Stokes orders having the highest peak power of ~300 W in the scheme with bulk AOM and the shortest duration of 5–7 ns in the scheme with fiber-pigtailed AOM. At the same time, the beam quality of generated pulses is greatly improved as compared to that for pump diode ($M^2 > 20$) reaching the best value ($M^2 = 2.05$) for the 2nd order Stokes beam in the scheme with the gain modulation and demonstrating also the most stable regime.

**Keywords:** fiber laser; Raman laser; multimode; graded-index; diode pumping; brightness enhancement; beam quality; mode locking



## 1. Introduction

The operating spectral region of fiber lasers can be significantly expanded exploring the stimulated Raman scattering (SRS) effect. With the help of SRS it is possible to obtain laser generation in the whole transparency window of silica-based passive fibers (1–2 μm) [1] at their pumping by conventional fiber lasers based on active fibers doped with rare-earth elements. Moreover, Raman fiber lasers (RFLs) may operate at wavelengths < 1 μm [2], where rare-earth-doped fiber lasers are either no longer able to generate, or have an extremely low quantum efficiency [3]. For a long time, attention was focused on Raman lasers built in a scheme with a singlemode fiber at relatively low pump power. The power density in the fiber core is sufficient to induce generation of Stokes wave via SRS gain and cavity feedback in a relatively short fiber [4]. In recent time, a new approach to Raman generation has been proposed: the use of multimode fibers with a graded (e.g., parabolic) refractive index profile of the core, which can be directly pumped by high-power multimode laser diodes [2,5,6], including development of an all-fiber scheme [7]. The highest pump-to-Stokes brightness enhancement factor of 73 has been obtained at conversion of highly multimode ($M^2 \sim 34$) pump radiation of LD at ~940 nm into a Stokes radiation at 976 nm with $M^2 < 2$ and power > 50 W in a graded-index (GRIN) fiber with 100-μm core [8,9]. Further operating wavelength range extension is possible via cascaded Raman conversion in GRIN fibers of multimode LD pump radiation into higher Stokes orders [10,11].

Pulsed operation of Raman lasers has some specific features associated with the instant interaction of pump and Stokes waves at their propagation in the long RFL cavity, which leads to a difference from the regimes of classical lasers based on short active fibers. The

first actively Q-switched Raman laser was demonstrated in a 1-km long singlemode (SM) fiber [12], where the formation of 1-μs pulses with an energy of 30 μJ was observed. Mode locking (ML) in Raman lasers based on SM fibers in all-fiber schemes was reported in active and hybrid (active-passive) ML regimes in [13,14], respectively. For multimode (MM) fibers, preliminary results on the active ML of Raman fiber laser were reported in [15].

Here we study different schemes of mode locked multimode Raman fiber lasers with direct diode pumping, including those with loss and gain modulation, and compare their output lasing characteristics.

## 2. Schemes of Pulsed Raman Lasers Based on Multimode Fiber

Three different laser schemes (Figure 1) are explored in the experiment. In all schemes, pump radiation of high-power multimode LD at wavelength of 976 nm is coupled (50% coupling efficiency) to a 62.5/125 multimode graded-index fiber with length of ~3.7 km. The first two schemes (Figure 1a,b) use LD in CW pumping regime and employ an acousto-optic modulator (AOM) to modulate the cavity losses. The RFL scheme with AOM has been first implemented in [12] for Q switching of SM RFL and similar one is proposed in [15] for mode locking of MM RFL. The difference between these two schemes lies in the bulk optics (AOM and end mirror with total insertion loss of ~10%) used in the first scheme (Figure 1a), while the second scheme (Figure 1b) has completely all-fiber cavity with fiber-pigtailed AOM. In the second case, the all-fiber cavity was formed by two fiber Bragg gratings (FBGs): UV inscribed FBG (R ~ 80%) in GRIN 62.5/125 fiber and femtosecond (fs) pulse inscribed FBG (R ~ 93%) (Figure 2a) in large-mode area (LMA) fiber with 10/125 μm core/cladding diameters. The fiber-pigtailed AOM with LMA 10/125 fiber ports is placed inside the fiber cavity between the MM GRIN Raman fiber and the fs-FBG. The LMA fiber with fs-FBG suppresses high-order modes, but at the same time, cavity losses become higher because of AOM insertion loss of ~2.3 dB. The third RFL scheme (Figure 1c) uses gain switching by means of LD current modulation thus eliminating limitations imposed by the AOM insertions. This cavity has also all-fiber configuration and was formed directly in GRIN fiber by two UV-inscribed FBGs with reflectivity R ~ 80% and R > 90% for output coupling (OC) and highly reflective (HR) cavity mirror, respectively. Dichroic mirrors M1, M2, M3 are used to separate pump radiation at 976 nm and generated Stokes radiation at ≥1018 nm, which is measured by optical spectrum analyzer (OSA) and photodetector (PD). In addition, some residual radiation reflected by M2 is used to measure profile of the generated beam by Thorlabs $M^2$-measurement system. Bandpass interference filter (IF) is used for the Stokes waves selection before measuring their beam quality with the $M^2$-meter.

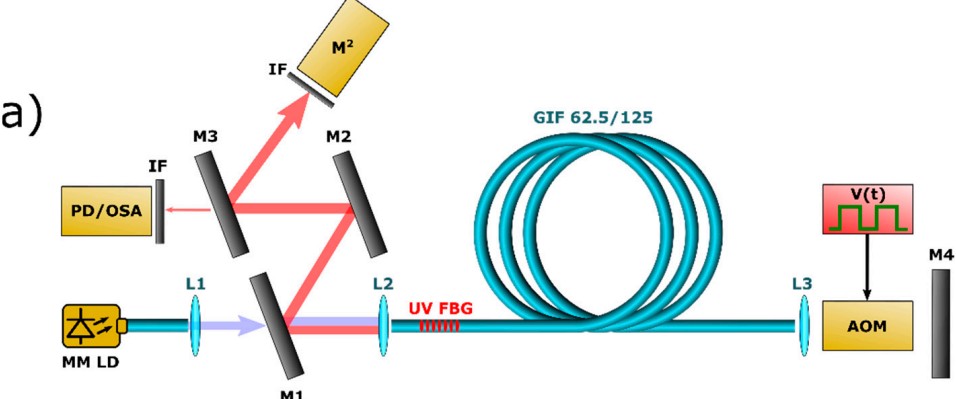

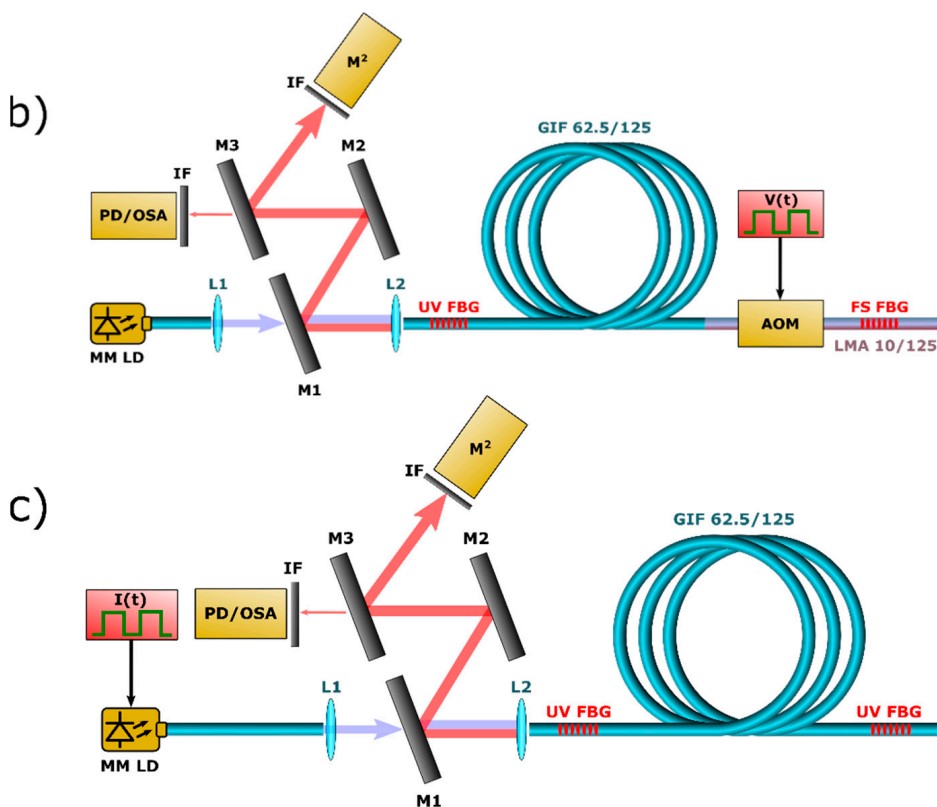

**Figure 1.** Experimental setup. Schemes with bulk AOM (**a**), LMA fiber-pigtailed AOM (**b**) and gain modulation (**c**): MM LD—high-power multimode laser diode at 976 nm; L1, L2, L3—collimating lenses; M1, M2, M3, M4—wavelength-selective dielectric mirrors; FBG—1018-nm fiber Bragg grating inscribed by UV or fs laser; MM GRIN fiber—multimode graded-index fiber; PD/OSA—photodetector or optical spectrum analyzer; $M^2$—$M^2$-meter; IF—bandpass interference filter, AOM—acousto-optic modulator.

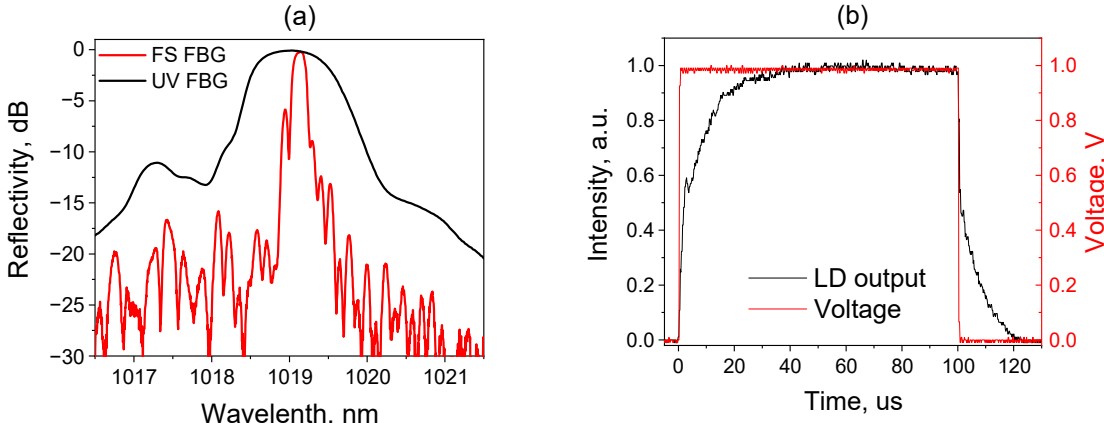

**Figure 2.** Reflection spectra of OC UV FBG and HR fs-inscribed FBG (**a**) and laser diode pump pulses and control electric square waveform signal used for gain modulation (**b**).

In all cases, the AOM or MM LD were driven by a square waveform electric signal with repetition rate that is nearly equal to the cavity intermode beating frequency of ~27 kHz (inverse roundtrip time) in order to obtain mode locking regime. Pulsed output signal is analyzed by a photodetector and an oscilloscope (Rigol DS6104) with bandwidths of 200 MHz and 1 GHz, correspondingly. It should be noted that the gain and loss switching times of these schemes determine the minimal width of pulse envelop. The AOM rise/fall time is of the order of tens of nanoseconds (20 ns for fiber-pigtailed AOM and ~100 ns for bulk AOM)

As for the gain switching technique, its LD current rise time is about 20 μs (see Figure 2b) and is determined mainly by the electric circuit. Since both the leading and the trailing edges have a considerable time duration, it was not possible to form a short pulse envelope in the gain modulation scheme.

In the next section we analyze and compare output characteristics of three different configurations with loss modulation by either bulk open-space or by fiber pigtailed AOM and with gain modulation without the use of AOM (Figures 1a, 1b and 1c, respectively). In all of the configurations the MM RFL may operate in mode locking regime, i.e., when switching period corresponds to the round-trip time of the pulse in the linear cavity.

## 3. Experimental Results

### 3.1. Open-Space Bulk AOM

When repetition rate of the AOM is set nearly equal to the signal round trip frequency (27.172 kHz) and input pump power (coupled to the GRIN fiber) reaches ~11 W, the 1st order Stokes generation at 1018 nm starts (Figure 3). At this moment, a sharp increase of the Stokes peak power is observed, but the pulse shape copies the AOM transmission window without significant shortening in this case [15], see Figure 4a, upper panel. At 14 W pump power, the second Stokes order at 1064 nm appears in the spectra and a multi-pulse generation mode with envelop of 400 ns is observed. With increasing pump power, individual sub-pulses become shorter (down to 14 ns), the time interval between them also shortens and the number of pulses increases until they completely fill the open AOM time window. The 2nd Stokes component was selected with a suitable band-pass filter and was registered by a photodetector. One can see that the 2nd Stokes multiple pulses became entirely separated, see Figure 4b. If we put the filtered 2nd Stokes sub-pulses on the total pulse envelope (red curve on the bottom panel of Figure 4a), we can clearly see that the total pulse envelope combines two components (1st and 2nd Stokes).

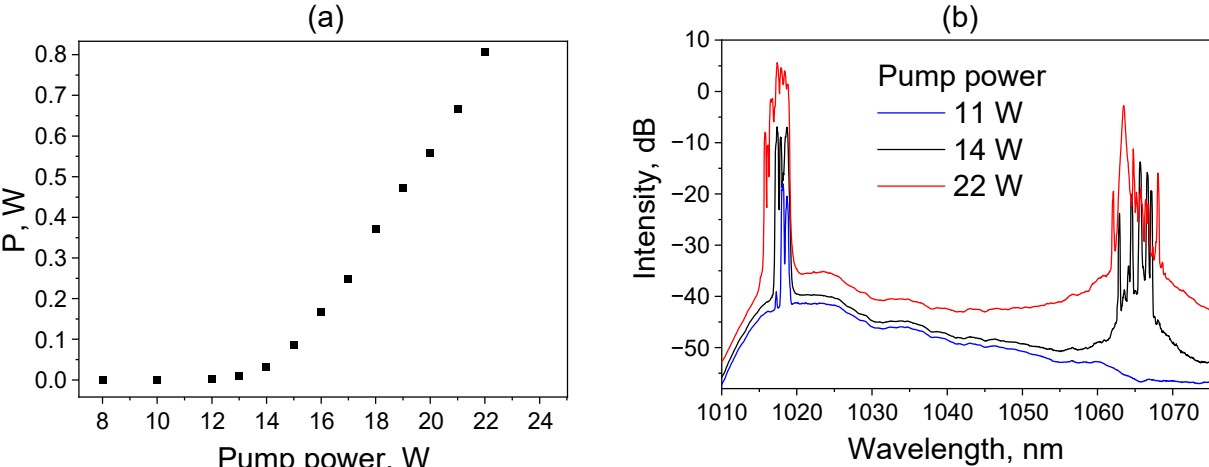

**Figure 3.** Output average power of RFL as a function of coupled to fiber input pump power (**a**) and output spectrum at different LD pump powers in 62.5-μm GRIN fiber Raman laser with the open-space AOM (**b**).

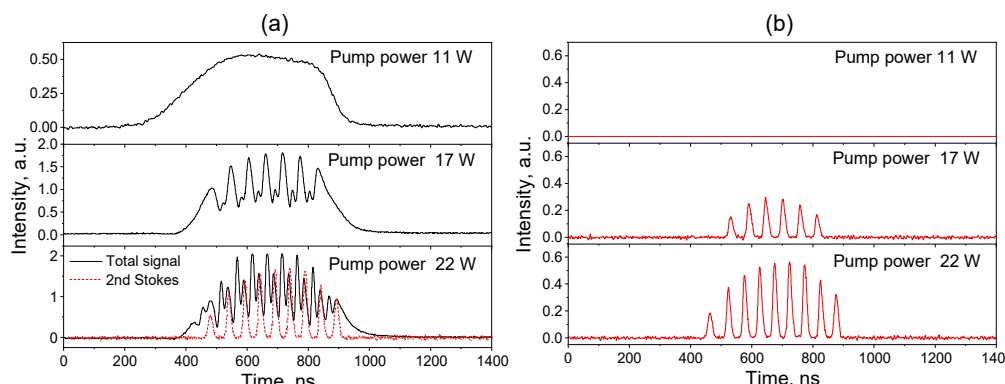

**Figure 4.** Output pulses (involving all Stokes orders) at different pump powers in 62.5-μm GRIN fiber Raman laser with open-space AOM (**a**) and filtered 2nd Stokes pulses (**b**).

Mechanisms of the multi-pulse regime in a mode-locked singlemode fiber Raman laser were studied in [14]. It was shown that the 2nd order Stokes generation begins to play a decisive role in the pulse shortening by depleting the 1st order Stokes pulse. In the presence of sufficient group velocity difference and excessive Raman gain along the pulse, several 2nd order pulses may be generated while the 1st Stokes pulse also becomes modulated within its envelope. In our experiments, the average output power of the signal containing both the 1st and the 2nd order Stokes waves reaches 810 mW at 22 W pump power. According to the optical spectra measured at 22 W pump power, the output signal contains about 50 mW of average power in the 2nd Stokes component. Taking into account the quantity of the 1st order Stokes sub-pulses (~7–8 pulses) and their duration of ~14 ns we could estimate the peak power of the sub-pulses with highest intensity as ~300 W.

Measurements of the RFL beam quality parameter $M^2$ at 20 W pump power give the value averaged over X- and Y-axes of ~5.3 for the 1st Stokes wave (Figure 5). The measured beam quality parameter for the 2nd Stokes beam is $M^2 \sim 3.3$, that exhibits more significant improvement compared to the pump beam quality ($M^2 \sim 26$).

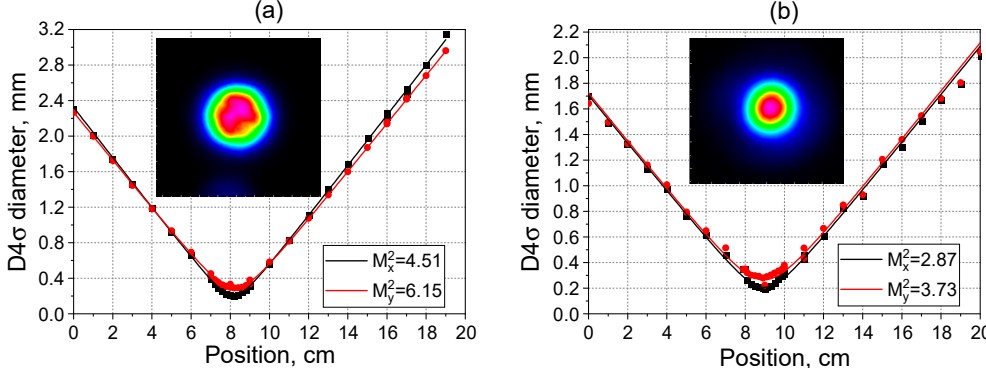

**Figure 5.** $M^2$ quality parameter measurements near the waist of the focused output beams of the RFL with open-space AOM: (**a**) for the generated 1st Stokes beam (1018 nm) at 20 W pump power and (**b**) for the 2nd Stokes beam (1064 nm) at 22 W pump power (~760 mW and 50 mW of the 1st and 2nd Stokes powers, respectively). Insets: corresponding intensity profiles near the beam waist.

### 3.2. LMA Fiber Pigtailed AOM

A rectangular shape pulse electrical signal with duration of 500 ns and repetition frequency of 27.19 kHz corresponding to the RFL cavity inverse round trip time is fed to the AOM in the second RFL configuration. At 32 W pump power the 1st order Stokes generation starts and the pulse shape copies the AOM transmission window. At 40 W pumping the 2nd Stokes order at 1064 nm appears in the spectra (Figure 6) and a multi-pulse generation mode with the envelope of 500 ns is observed (Figure 7), on both sides of the cavity. Finally, at 44.4 W pump power the continuous generation appears between

the pulses. Although we cleaved the fiber ends at an angle of ~10° in order to reduce feedback associated with Fresnel reflection, the distributed feedback provided by Rayleigh backscattering in the 3.7-km long fiber becomes sufficient for continues cascaded generation of the 1st and 2nd Stokes waves at such high pump power. Note that similar random lasing was observed in a long 62.5-µm GRIN fiber with 940-nm LD pumping at the absence of AOM [16].

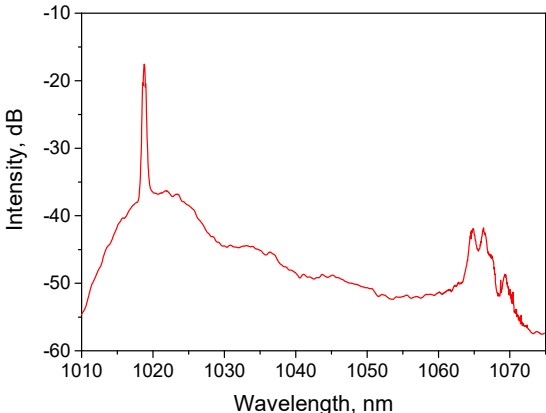

**Figure 6.** Output spectrum (40 W pump power) in 62.5-µm GRIN fiber Raman laser with fiber pigtailed AOM.

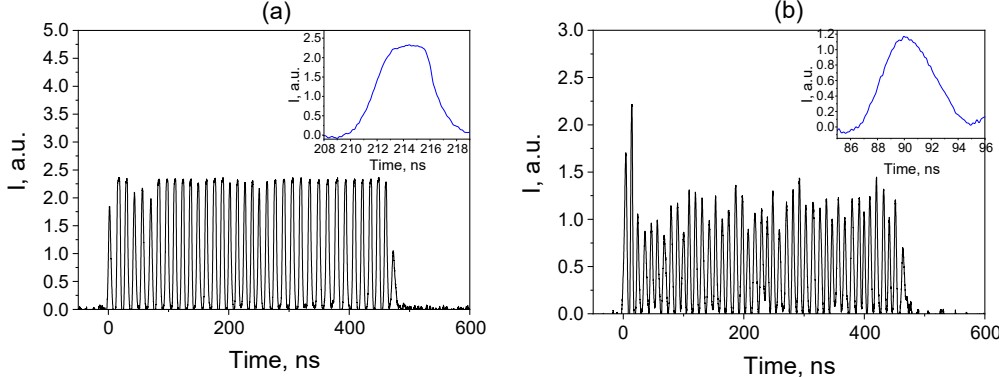

**Figure 7.** Output pulse shapes of the 1st Stokes (1018 nm) (**a**) and the 2nd Stokes (1064 nm) (**b**) waves at 40 W pump power in the RFL with fiber pigtailed AOM. Insets: individual sub-pulse shapes.

In the mode-locking regime, the average power of the 1st Stokes wave amounts to about 30 mW for this configuration. Taking into account the number of the 1st order Stokes sub-pulses within the envelope (Figure 7a) and individual pulse duration of about 6–7 ns we could estimate the peak power of the shortest pulses as ~10 W. It should be noted that the 2nd order Stokes pulse contains sub-pulses with slightly shorter duration (5–6 ns), (Figure 7b), but their envelope is less uniform and less stable.

Measurement of the RFL beam quality at 40 W pump power gives $M^2 = 2.86$ for the 1st Stokes wave (Figure 8) that is significantly better in comparison with the scheme, utilizing an open-space bulk AOM. The measured quality parameter of the 2nd Stokes beam is $M^2 = 2.08$ that is also sufficiently improved compared to the 1st Stokes beam. At the same time, the measured spectra (Figure 6) show that the 2nd Stokes spectrum generated without FBG cavity is much broader than the 1st Stokes one. Note that the latter is defined by the narrowband fs-FBG inscribed in the LMA fiber, see Figure 2a.

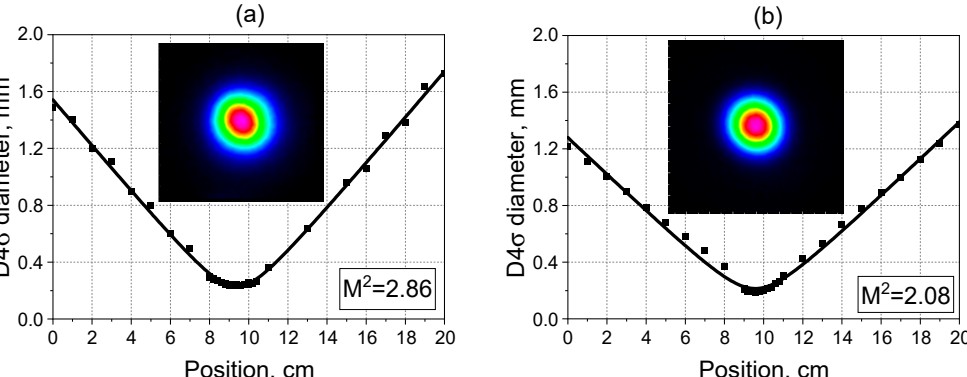

**Figure 8.** Measured beam quality parameter $M^2$ of the 1st Stokes (1018 nm) (**a**) and the 2nd Stokes (1064 nm) (**b**) beams at 40 W pump power. Insets: corresponding intensity profiles near the beam waist.

### 3.3. Gain Modulation

In the scheme with gain modulation, a pulsed current with a rectangular shape and a pulse repetition frequency close to the cavity inverse round-trip time (~27.2 kHz) was applied to the laser diode. As the existing current driver does not allow us to generate short current pulses, the pulse duration is as long as 5 μs in this case. Similar to previous schemes, multiple periodic pulses are observed within the envelope of the output signal, when the gain modulation frequency is tuned close to the mode locking resonance and the generation threshold of the 2nd Stokes generation is reached.

Figure 9 shows the generated pulses at the laser output at different modulation frequencies of the pump laser diode. It can be seen that at a frequency of 26.5 kHz, a long pulse (1 μs) is formed at the trailing edge, which shortens with increasing frequency and additional peaks appear at the same time. Near the resonant frequency (27.2 kHz), the resulting pulse is modulated and consists of many sharp peaks with duration of about 15 ns. At a further increase in the LD current modulation frequency, the sub-pulses become again aperiodic and long. Note that the output pulse shape is quite stable in time, in spite of that the pulse envelope takes the form of "noisy" pattern with random-like structure of sub-pulses (for example, at 27.7 kHz). The specific structure of the generated pattern (quantity of sub-pulses, their individual durations, relative positions inside the envelope, jitter, etc.) is stable and doesn't change over several minutes while the laser is operating. Only individual amplitudes of sub-pulses are slightly varying from pulse to pulse due to thermal processes inside the long cavity.

Figure 10 shows output spectrum generated at a repetition rate of 27.2 kHz and a pump power of 40 W. The spectral line widths at −3 dB level for the first and second Stokes generation were 0.4 and 0.2 nm, respectively. At the same time, the generation line has a broader background with spikes at the spectral tails, see inset in Figure 10. Figure 11 shows the spectrally separated 1st and 2nd Stokes pulses near the resonant frequency for a pump pulse duration of 8 μs. It can be seen that the 1st Stokes pulse with a duration of 3 μs is formed, however, the 2nd Stokes generation occurs only at its pulse maximum, where modulation is observed. The 2nd Stokes pulse shape with 100% intra-pulse modulation appears with an envelope profile of ~1.5 μs duration (Figure 11b). In this regime, the individual pulse duration is about 15 ns and 12 ns for the 1st and the 2nd Stokes components, respectively. The corresponding average powers of the 1st and 2nd Stokes generation were about 100 mW (~2.7 W peak power) and 20 mW (~1 W peak power), respectively. In addition, the quality parameter $M^2$ for the 1st and 2nd Stokes beams is measured to be 3.29 and 2.05, respectively (Figure 12). So, in the gain-modulated scheme the 2nd Stokes beam has the best beam quality. Let us also note that in this scheme, the spatial mode structure is more stable over time compared to the previous schemes with AOM described above, and it is maintained for at least several minutes without adjusting the repetition rate. This is mostly could be reasoned by the lower thermal load of the

fiber due to pulsed pumping and, as a consequence, the smaller relative variations of the resonator round trip time.

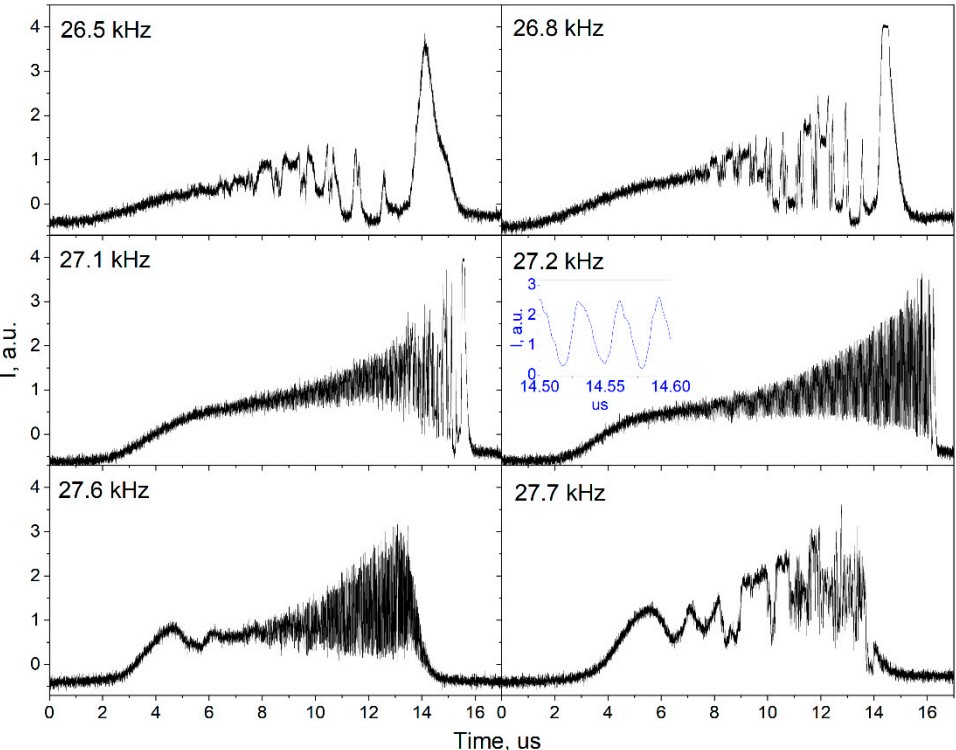

**Figure 9.** Output pulses at different repetition rates of the LD current modulation. Inset: zoomed sub-pulses.

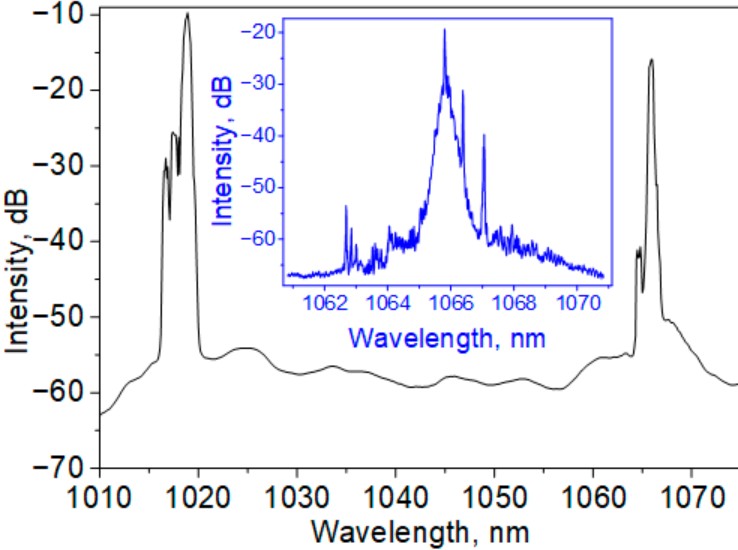

**Figure 10.** Output spectrum (40 W pump power) in 62.5-μm GRIN fiber Raman laser with gain modulation. Inset: zoomed 2nd Stokes spectrum with better resolution.

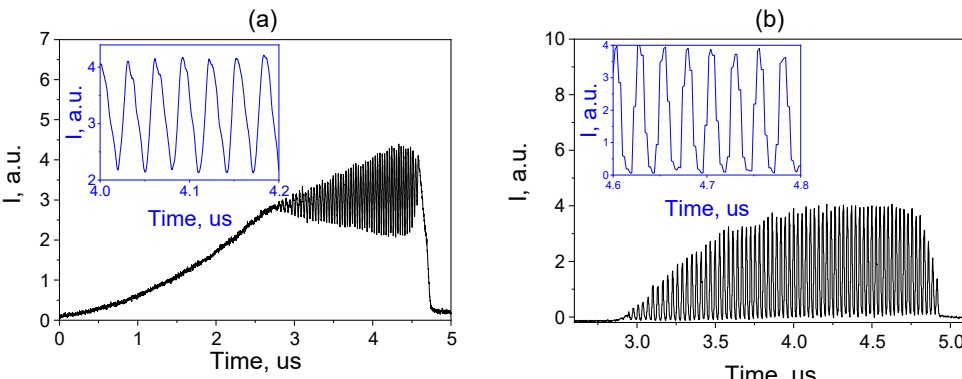

**Figure 11.** The 1st Stokes (**a**) and the 2nd Stokes (**b**) pulse shapes at gain modulation (40 W pump power, repetition rate 27.22 kHz). Inset: zoomed sub-pulses.

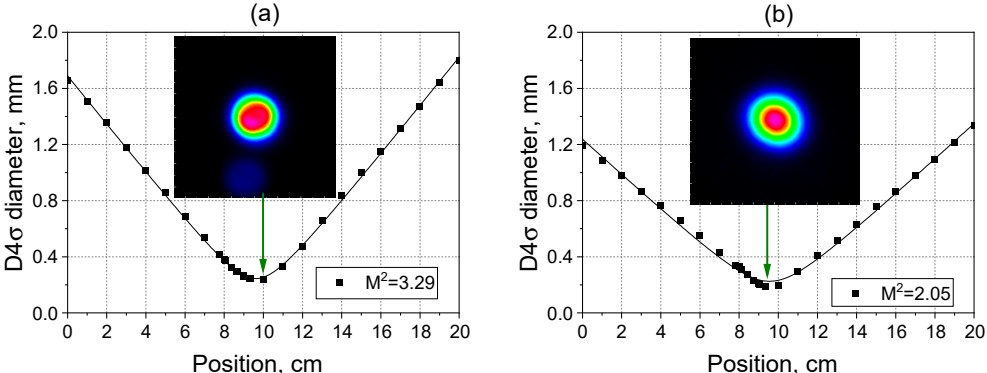

**Figure 12.** Measured quality parameter $M^2$ of the 1st Stokes (1018 nm) (**a**) and the 2nd Stokes (1064 nm) (**b**) beams at 40 W pump power. Insets: corresponding intensity profiles near the beam waist.

## 4. Discussion and Conclusions

Thus, three different schemes of a pulsed Raman laser based on a multimode GRIN fiber pumped by high-power MM LD were studied in this work. In all three schemes, when the threshold of the 2nd Stokes generation is exceeded, the output pulses take the form of a train of short incoherent nanosecond sub-pulses for both the 1st and 2nd Stokes with an envelope that repeats the AOM transmission or the gain switching (light emission of the LD pump) profiles. The incoherent noise-like structure of pulses is typical for km-long mode-locked fiber lasers [17]. Since the lengths of the cavities are approximately the same (3.7 km) in all our schemes, the resonant frequency is also the same amounting to about 27.2 kHz. The variation of output pulses with modulation frequency is very similar to the case of Yb fiber laser with quasi-synchronous modulation of pump power [18]. Despite the obvious similarities of the multi-pulse phenomenon, the nature of the formation of sub-pulses may be different and in our case, it is defined by the interaction of three waves [14]: pump, 1st and 2nd Stokes. For example, when placing the normal cleaved output end (Figure 1b,c) in glycerin, the generation of the 2nd Stokes wave disappears, as well as the multi-pulse pattern of the output signal. Note that the regular structure of the 1st and 2nd Stokes multi-pulse patterns is formed near the round-trip resonance (but not exactly, that may be necessary for compensation of group velocity dispersion between the 1st and 2nd Stokes pulses). At large detuning from the resonance, the dispersion may lead to the observed irregularities in the both patterns, whereas their random-like shapes are stable in time. So, by varying detuning we can generate multi-pulse patterns of different shapes that may be treated as an alternative to active shaping of the pulse bursts, see, e.g., [19], demanded for specific applications in material processing.

Comparing different schemes, we can conclude that for open-space AOM the resonator has relatively low losses, and, moreover, the unabsorbed pump beam after the AOM is fed back into the cavity, increasing the SRS gain. In this case, the maximum output power of SRS Stokes was obtained (810 mW total power with 400 ns envelope), but the beam quality turned out to be the lowest among three schemes ($M^2 \sim 5$ and $M^2 \sim 3$ for 1st and 2nd Stokes, respectively). All investigated schemes generate pulses that are longer compared to the SMF mode locked Raman laser (~300 ps in a 500 m cavity) [14] and this can be partially explained by influence of intermodal dispersion in MM fiber and its long cavity length.

The all-fiber laser scheme using a pigtailed fiber AOM allows modulating losses with a minimum response time, and due to this, rectangular pulse shape with almost 100% intra-pulse modulation were obtained for both the 1st and 2nd Stokes order generation. However, due to the mismatch of the GRIN fiber and LMA fiber in which the fs-FBG with narrow reflection spectrum is formed, both leading to relatively high losses, the total average power was only ~30 mW (with a 500 ns AOM opening time). However, the individual pulse duration is shortened to 5–7 ns and the quality of 1st and 2nd order Stokes beams is improved to $M^2 \sim 2.86$ and 2.08, respectively. Both schemes with AOMs use continuous pumping, and due to the heating of the entire cavity, the stability of the regime is not good being perturbed in ~10 s, which requires adjustment of the AOM modulation frequency.

Implementation of the circuit with gain switching, where the LD current and the pump power were modulated, respectively, turned out to avoid this drawback, and the stability of the regime has been increased significantly. However, the home-made electrical circuit for controlling the LD current did not allow the LD to be switched on quickly, and therefore rather long pulses (8–16 µs) were used for pumping. In this case, the 1st Stokes generation occurred only at the top of the pump pulse, whereas the 2nd Stokes pulse shape with 100% intra-pulse modulation was observed. We also managed to achieve beam quality of $M^2 \sim 3.3$ and 2.05 for the first and second order Stokes beams, respectively, in spite of the absence of fs-FBG in LMA fiber with spatial filtering properties that indicates the influence of nonlinear effects such as Kerr cleaning [20] which is shown to be important in the scheme of CW multimode fiber Raman laser with LD pumping of comparable power [21]. It should be noted that the average power amounting to 120 mW is less than that for open-space AOM. One of the reasons is a narrower beam profile consisting of low order modes as indicated by ~2 times smaller $M^2$ value that results in lower power at the same intensity. The maximum peak power of the 1st Stokes beam was reached in the scheme with an open-space AOM (~300 W) because it has lowest losses and double-pass of pump wave. The scheme with a fiber-pigtailed AOM has highest losses and ~10 W peak power of the 1st Stokes beam. The third scheme with gain modulation has only 2.7 W peak power mainly because of non-optimal pump modulation. Corresponding 1st Stokes pulse energies for three different schemes are 4.2 µJ, 60 nJ and 41 nJ, respectively. At the same time, both the spatial and temporal structure of Raman generation in the last case is the most stable among three schemes. Although the duration of individual pulses (~12 ns) is not the shortest it may be further shortened as the technical task of creating a current driver with a short response time is feasible. So, the laser scheme with gain modulation looks the most promising, because it has better stability, fewer optical components, and good output beam quality.

Thus, the developed LD-pumped Raman fiber lasers (RFLs) are able to generate kHz-range trains of microsecond bunches of nanosecond pulses with peak power up to 0.3 kW. The generated wavelength range in the presented proof-of-principle experiments is the same as that for mode-locked Yb-doped fiber lasers (YDFLs) generating either noise-like [17,18] or coherent pulses [22,23]. At the same time, the operating range of RFLs may be shifted to shorter wavelengths, <980 nm (if high-power LDs with wavelengths 915–940 nm or ~800 nm will be applied for pumping), where operation of YDFLs is not possible. A shift to longer wavelength is also feasible if higher Stokes orders will be explored. In practical applications, this type of lasers can be useful for micromachining, LIDAR, laser cleaning, biomedicine, sensing etc.

**Author Contributions:** Conceptualization, S.A.B.; investigation, A.G.K., S.I.K. and Y.A.T. All authors have read and agreed to the published version of the manuscript.

**Funding:** The study is supported by Russian Foundation for Basic Research (21-72-30024).

**Acknowledgments:** The authors acknowledge FBG fabrication by I. Nemov and A. Wolf.

**Conflicts of Interest:** The authors declare no conflict of interest.

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
