# Peer review of "Actively Mode Locked Raman Fiber Laser with Multimode LD Pumping"

_photonics, doi:10.3390/photonics9080539_

Round 1

Reviewer 1 Report

In my opinion it’s questionable to regard the demonstrated lasers as “mode-locked laser”, and it’s more appropriate to call them as “Q-switched” and “gain-switched” lasers. The authors may argue that in the Q-switched or gain-switched temporal profile, there are multi-pulses with a specific period, which even corresponds to the roundtrip time of the cavity. However, the mechanism of the multi-pulse operation was actually not discussed in the manuscript, only in the conclusion section the authors mentioned that those pulses are noise-like pulses, which are pulses bunches that comprised with random pulses, and cannot be leveraged to verify the mode-locking operation.

Reviewer 2 Report

Thank you for your article. It is well written and can be useful for a practical and scientific point of view.

For better understanding and ability to repeat your results, it is good to clarify next points:

1) Fig. 1 a. Losses in the right part of the scheme were not discussed (insertion loss of AOM, coupling efficiency  between fiber end and AOM and AOM and mirror).

2) Please, provide loss estimation in splice connection between GRIN and LMA. AOM insertion losses are also missed in this case.

3) Did you observe radiation from another side of your cavities (2nd Stokes?)? 

Please, check your work on self-plagiarism.

1) Results from part 3.1 Open-space bulk AOM are very close to your previous publication doi: 10.1117/12.2537769 (fig. 5 = fig. 4 (A) of current work, fig. 6 (A) = fig. 4 (B) of current work)

Reviewer 3 Report

In this work, Kuznetsov et.al. proposed the pulsed regimes of Raman conversion of highly multimode laser diode pump radiation into the 1st and higher order Stokes radiation in multimode graded-index fibers. Three different linear cavities of Raman fiber laser with the modulation 10 of losses are explored and compared. The results presented are of interest to the laser community and the topic is within the scope of Photonics, however I do not believe this work is impactful enough for Photonics and nor is the technical content high enough in the present form. I therefore recommend this paper be reconsidered after major revision. 

1.         Pulsed operation of 1um fiber laser has been well developed. In this paper, the wavelength of 1018nm and 1064nm pulsed lasers were obtained by the Raman conversion of multimode LD pump radiation into higher Stokes orders. The comparison with the latest results of rare-earth-doped fiber lasers should be updated in the Introduction.

2.         In the abstract, the authors claimed that “nanosecond pulses have been observed for both Stokes orders having the highest peak power of ~300W in the scheme with bulk AOM and the shortest 15 duration of ~5 ns in the scheme with fiber-pigtailed AOM”. However, there are no relevant results in the manuscript. And for the pulsed laser experiments, the pulse profile, frequency spectrum as well as the output power should be presented.

3.         In “Figure.7” and “Figure.11.”, the pulse trains seem unstable. This needs to be well-explained.

Round 2

Reviewer 3 Report

The manuscript can be accepted except one question below. I still recommend the authors add more details about the "looks quite noisy but structure stable in time" pulses, how can this phenemonon happen and are there any references about this.
